# ON STATIONARY POINT CONVERGENCE OF PPO-CLIP

**Ruinan Jin**
The Chinese University of Hong Kong, Shenzhen
`jinruinan@cuhk.edu.cn`

**Shuai Li**
Shanghai Jiao Tong University
`shuaili8@sjtu.edu.cn`

**Baoxiang Wang**[*]
The Chinese University of Hong Kong, Shenzhen
`bxiangwang@cuhk.edu.cn`

## ABSTRACT

Proximal policy optimization (PPO) has gained popularity in reinforcement learning (RL). Its PPO-Clip variant is one the most frequently implemented algorithms and is one of the first-to-try algorithms in RL tasks. This variant uses a clipped surrogate objective function not typically found in other algorithms. Many works have demonstrated the practical performance of PPO-Clip, but the theoretical understanding of it is limited to specific settings. In this work, we provide a comprehensive analysis that shows the stationary point convergence of PPO-Clip and the convergence rate thereof. Our analysis is new and overcomes many challenges, including the non-smooth nature of the clip operator, the potentially unbounded score function, and the involvement of the ratio of two stochastic policies. Our results and techniques might share new insights into PPO-Clip.

## 1 INTRODUCTION

Proximal policy gradient (PPO) is one of the most popular algorithms in reinforcement learning (RL) (Schulman et al., 2017). For many RL tasks, one would try PPO as their first attempt, for its generality and stability among different environments as well as the easiness of implementation. As of September 2023, it returns over 5,500 results on GitHub.com, if one searches "PPO". This demonstrates enthusiasm for reproducing the algorithm, applying it to different problems, and extending it to many variants.

PPO stems from policy gradient, which is proposed by Williams (1992) as one of the seminal works in RL. The idea is to utilize the policy in a previous step as an off-policy reference, and restrain the new policy to be within a small discrepancy of the old policy. In this way, the policy does not suddenly deviate from its previous iterations by too much. The idea is well-compatible with modern deep learning models, such as advantage function estimation and policy parametrization.

There are two primary variants of PPO introduced by Schulman et al. (2017). For the sake of convenience, we call them PPO-Penalty and PPO-Clip. PPO-Penalty approximately solves a KL-constrained update, but penalizes the KL-divergence in the objective function instead of making it a hard constraint. The penalty coefficient is then automatically adapted over the course of the training so that it scales appropriately. PPO-Clip does not have a KL-divergence term in the objective and does not have a constraint either. Instead, it relies on a clipping operator in the objective function to remove incentives for the new policy to get too far away from the old policy. This use of a clipped surrogate objective function is not typical in optimization algorithms. In this work, we focus on PPO-Clip.

In contrast to the empirical success of PPO-Clip, theoretical analysis of it is limited. Huang et al. (2021) and Yao et al. (2022) have made substantial attempts to discuss the convergence of PPO-Clip under the tabular and the over-parameterized neural network settings. The basic idea is to categorize PPO-Clip as a hinge loss problem in classification. This analysis relies on the local convexity of the optimization objective, with which typical techniques for hinge loss apply. Liu et al. (2019)

---
[*]Corresponding to: Baoxiang Wang

employed the classic proof based on the neural tangent kernel (NTK) condition to establish the global convergence of over-parameterized neural networks. Their analysis is for a variant of PPO which no longer involves the ratio of the current policy and the old policy.

In fact, analyzing the original PPO-Clip involves many challenges. First, the algorithm uses the ratio of two stochastic policies, which is a unique structure that has no off-the-shelf recursive inequality available to characterize it. Second, the clip operator is non-smooth, which necessitates the use of divide-and-conquer of probabilistic events. In comparison, treating the clip operation as a hinge loss does not provide more help when the problem is general and not locally convex. Third, the score function of the policy is unbounded. A typical example is the commonly used neural network policy. Without bounded score function, strong conditions such as the gradient dominance lemma are no longer available, and existing techniques in analysis of policy gradient will not apply. In addition, the unbounded score function could make the ratio of two policies arbitrarily large, even in the late stages of the optimization process.

In this work, we investigate the vanilla version of PPO-Clip with a minimum set of conditions. We formulate PPO-Clip into a general two-scale iterative process, where the old policy $\pi_{\text{old}}$ is synchronized every $T$ steps ($T$ could be 1, which makes it single-scale). We show that this process will converge to a stationary point of the learning objective, up to a bias term that depends only on the biases incurred by the Markovian sampling and the advantage estimation. When the biases are zero, such as when we use Monte-Carlo methods, PPO-Clip converges almost surely. We further provide the convergence rate of the gradient norm in terms of average-iterate convergence and subsequence convergence. Technically, our analyses are based on key lemmas that characterize the probabilities of the clip events and thereof the recursion of the learning process. This new tool might share insights into analyzing PPO and might be of independent interest.

## 2 PRELIMINARIES

### 2.1 MARKOV DECISION PROCESS

We consider a canonical discounted Markov decision process (MDP) setting with finite state space and action space. In this work, an MDP is denoted as $\mathcal{M} = (\mathcal{S}, \mathcal{A}, P, r, \gamma)$, where $\mathcal{S}$ and $\mathcal{A}$ are finite state and action spaces, $P : \mathcal{S} \times \mathcal{A} \to \Delta(\mathcal{S})$ denotes the transition kernel, $r : \mathcal{S} \times \mathcal{A} \to \mathbb{R}$ is the reward function, and $\gamma \in (0, 1)$ is the discount factor. Let $s_0$ be the initial state.

The goal of reinforcement learning is to learn a policy $\pi$ that takes $s \in \mathcal{S}$ and outputs a distribution $\pi(\cdot|s)$ over the action space, to maximize the expected cumulative discounted reward. For every policy $\pi$, define the action value function $Q^\pi : \mathcal{S} \times \mathcal{A} \to \mathbb{R}$ as $Q^\pi(s, a) := \mathbb{E}_{\pi, P}\left(\sum_{h=0}^{\infty} \gamma^h r(s_h, a_h) \middle| s_0 = s, a_0 = a\right)$. Here, $\mathbb{E}_{\pi, P}(\cdot)$ denotes the expectation under subsequent actions and states that follow $\pi$ and $P$. When the context is clear we write this into $\mathbb{E}_\pi(\cdot)$ instead. Define the state-value function $V^\pi : \mathcal{S} \to \mathbb{R}$ as $V^\pi(s) := \mathbb{E}_{a \sim \pi(\cdot|s)}[Q^\pi(s, a)]$ and the advantage function $A^\pi : \mathcal{S} \times \mathcal{A} \to \mathbb{R}$ as $A^\pi(s, a) := Q^\pi(s, a) - V^\pi(s)$, respectively. In this paper, we consider the case where the policy is parametrized by a parameter $\theta \in \mathbb{R}^d$. Define, as the objective function of our optimization,

$$V(\theta) := V^{\pi_\theta}(s_0) = \mathbb{E}_{\pi_\theta}\left(\sum_{h=0}^{+\infty} \gamma^h r(s_h, a_h) \middle| s_0\right).$$

Denote $V^* := \max_{\theta \in \mathbb{R}^d} V(\theta)$ as a global maximum of $V(\theta)$.

### 2.2 POLICY GRADIENT

When both $\pi$ and $V$ are parametrized by $\theta$, it is natural to consider gradient ascent methods to optimize the objective function, i.e., $\theta_{n+1,1} = \theta_{n,1} + \epsilon_n \nabla V(\theta_{n,1})$, where $\epsilon_n > 0$ is the learning rate (step size). The gradient term $\nabla V(\theta)$ can be estimated via $\nabla V(\theta) = \mathbb{E}_{\pi_\theta}\left(\nabla \ln \pi_{\theta_n}(a|s) \tilde{A}^{\pi_\theta}(s, a)\right)$, where $\tilde{A}^{\pi_\theta}$ is an estimation of the advantage function $A^{\pi_\theta}$ (Williams, 1992; Sutton & Barto, 2018). The policy gradient (PG) estimation is unbiased whenever $\tilde{A}$ is an unbiased estimation of $A$, for example, the Monte-Carlo estimation.

Policy gradient have received a significant line of research and improvement, and together with the emergence of deep neural networks, receives empirical success in many areas (Konda & Tsitsiklis, 2000; Kakade, 2001; Schulman et al., 2017; 2015; Lillicrap et al., 2015; Schulman et al., 2017). In theory, the global convergence of PG is investigated under the gradient dominance condition (Zhang et al., 2020; 2021; Yuan et al., 2020; Xu et al., 2020; Huang et al., 2020; Fazel et al., 2018; Fatkhullin et al., 2023) and soft-max policy condition (Agarwal et al., 2021; Mei et al., 2020). Stationary point convergence of PG under general policies are proved by Zhang et al. (2020); Fazel et al. (2018). Wang et al. (2019) study the convergence of neural policy gradient methods. While our setting is quite different to PG, we obtain some insights from these lines of theoretical works.

## 2.3 Proximal Policy Optimization with Clipped Surrogate Objective

The proximal policy optimization algorithm has different variants (Schulman et al., 2017; Huang et al., 2022; Hsu et al., 2020; Zhang et al., 2019; 2022). In this work, we are particularly interested in investigating its variant with a clipped surrogate objective which is introduced in the original PPO work. We denote this variant as PPO-Clip in our manuscript. The surrogate objective function is defined as $L^{\text{CPI}}(\theta) = \mathbb{E}_{\pi_{\theta_{\text{old}}}} \left( \frac{\pi_\theta(a|s)}{\pi_{\theta_{\text{old}}}(a|s)} \tilde{A}_{\pi_{\theta_{\text{old}}}(s,a)} \right)$, where $\pi_{\theta_{\text{old}}}$ denotes the policy at a previous iteration, $\tilde{A}^{\pi_\theta}$ is an estimation of the advantage function $A^{\pi_\theta}$. The surrogate objective function with the clip operator is defined as $L^{\text{CLIP}} = \mathbb{E}_{\pi_{\theta_{\text{old}}}} \left( \min\left\{ \frac{\pi_\theta(a|s)}{\pi_{\theta_{\text{old}}}(a|s)}, \text{clip}\left( \frac{\pi_\theta(a|s)}{\pi_{\theta_{\text{old}}}(a|s)} \right) \right\} \tilde{A}_{\pi_{\theta_{\text{old}}}}(s, a) \right)$. The first term in the minimum operator is $L^{CPI}$. The second term, $\text{clip}\left( \frac{\pi_\theta(a|s)}{\pi_{\theta_{\text{old}}}(a|s)} \right) \tilde{A}_{\pi_{\theta_{\text{old}}}}(s, a)$, modifies the surrogate objective by clipping the probability ratio to the interval $[1 - \delta_0, 1 + \delta_0]$, for some absolute constant $\delta_0$. PPO-Clip takes the minimum of the clipped and unclipped objective and the final objective is a lower bound on the unclipped objective.

In the original PPO work, PPO-Clip considers $\pi_{\theta_{\text{old}}}$ to be the policy used in the immediate previous iteration. In the current iteration, the algorithm aims to find a parameter $\theta$ that maximizes the clipped surrogate objective, using some estimation $\tilde{\mathbb{E}}_{\pi_{\theta_{\text{old}}}}$ of the expectation $\mathbb{E}_{\pi_{\theta_{\text{old}}}}$ over the Markovian sampling. Namely,

$$\text{maximize}_{\theta \in \mathbb{R}^d} \tilde{\mathbb{E}}_{\pi_{\theta_{\text{old}}}} \left( \min\left\{ \frac{\pi_\theta(a|s)}{\pi_{\theta_{\text{old}}}(a|s)}, \text{clip}\left( \frac{\pi_\theta(a|s)}{\pi_{\theta_{\text{old}}}(a|s)} \right) \right\} \tilde{A}_{\pi_{\theta_{\text{old}}}}(s, a) \right). \tag{1}$$

This maximization alone could involve multiple steps of updates on $\theta$, forming a double-loop structure of the PPO algorithm. The off-policy nature of the surrogate objective specifically fits into this type of double-loop structure. In practice, such a structure is implemented simply by running the optimizer multiple times (e.g., *optimizer.step()* in PyTorch) for each policy gradient update.

## 3 Setting and Results

In this section, we aim to prove the global convergence of PPO-Clip to a stationary point. Namely, the gradient of the value function $V(\theta)$ should converge to zero whenever the estimation bias is zero (Theorem 3.1 and 3.2). We further extend this result to the convergence rate of average-iteration convergence and subsequence convergence (Theorem 3.3).

### 3.1 Formulation

Now we are ready to formulate the PPO-Clip variant that we investigate in this paper. In this work, we specifically implement the optimization process within one step of policy gradient, namely Equation (1), as $T$ steps of gradient ascent. The algorithm therefore runs in a two-scale manner, where the index $n$ of policy gradient iteration will be considered asymptotically large, and the index $k$ of the maximization in (1) is considered to be within $1, \ldots, T$, for some small constant $T$. Notice that all our results hold for $T = 1$ as well. We formulate PPO-Clip as below.

**PPO-Clip:**

$$\theta_{n,k+1} = \theta_{n,k} + \epsilon_{n,k} \nabla_{\theta_{n,k}} \left( \tilde{\mathbb{E}}_{\pi_{\theta_{n,1}}} \left( \min \left\{ \mathrm{clip}\left( \frac{\pi_{\theta_{n,k}}(a|s)}{\pi_{\theta_{n,1}}(a|s)} \right), \frac{\pi_{\theta_{n,k}}(a|s)}{\pi_{\theta_{n,1}}(a|s)} \right\} \cdot \tilde{A}_{\pi_{\theta_{n,1}}}(s,a) \right) \right),$$
$$(k = 1, 2, ..., T-1),$$
$$\theta_{n+1,1} = \theta_{n,T},$$

$$(2)$$

where $\tilde{\mathbb{E}}_{\pi_{\theta_{n,1}}}$ is an estimate of $\mathbb{E}_{\pi_{\theta_{n,1}}}$, and $\tilde{A}_{\pi_\theta}(s,a)$ is the estimated value of the advantage function.

As we have mentioned in the preliminary section on PPO, the value of $A$ and the expectation under Markovian sampling $\mathbb{E}_{\pi_{\theta_{n,1}}}$ need to be estimated. Both the estimations can be obtained in an unbiased way using Monte-Carlo sampling, provided that the sampling goes through until the termination of the episode. Practically, PPO is rarely implemented in a pure Monte-Carlo way. Methods such as the actor-critic algorithm use biased estimations to trade for other properties such as sample complexity and generalization. Nevertheless, this biasedness can be formulated by the sampling bias term.

Define the functionals

$$\mathbb{E}^\circ_{\pi_\theta}(\cdot) := \mathbb{E}_{\substack{s_0\sim\rho,\ a_h\sim\pi_\theta(\cdot|s_h) \\ s_{h+1}\sim\mathbb{P}(\cdot|s_h,a_h)}} \left( \cdot \times A_{\pi_\theta}(s,a) \right), \ \ \tilde{\mathbb{E}}^\circ_{\pi_\theta}(\cdot) := \tilde{\mathbb{E}}_{\substack{s_0\sim\rho,\ a_h\sim\pi_\theta(\cdot|s_h) \\ s_{h+1}\sim\mathbb{P}(\cdot|s_h,a_h)}} \left( \cdot \times \tilde{A}_{\pi_\theta}(s,a) \right).$$

If $\tilde{A}_{\pi_\theta}$ is bounded, for which we will assume in Assumption 3.2, both of the functionals are bounded linear functionals in $\mathcal{S} \times \mathcal{A} \to \mathbb{R}$. Therefore, $(\mathbb{E}^\circ_{\pi_\theta} - \tilde{\mathbb{E}}^\circ_{\pi_\theta})(\cdot)$ is also a bounded linear functional. The norm of this functional then exists and we denote the norm as $\left\| (\mathbb{E}^\circ_{\pi_\theta} - \tilde{\mathbb{E}}^\circ_{\pi_\theta})(\cdot) \right\| = \max_{\|f\|=1} \left| (\mathbb{E}^\circ_{\pi_\theta} - \tilde{\mathbb{E}}^\circ_{\pi_\theta})(f) \right|$, where $\|f\| := \max_{(s,a)\in\mathcal{S}\times\mathcal{A}} |f(s,a)|$. Subsequently, denote the upper bound of the norm of this functional as

$$\left\| (\mathbb{E}^\circ_{\pi_\theta} - \tilde{\mathbb{E}}^\circ_{\pi_\theta})(\cdot) \right\| \leq \phi_n. \tag{3}$$

When Monte-Carlo sampling is used to estimate the expectation $\tilde{\mathbb{E}}$ of the Markovian sampling and the advantage function $\tilde{A}$, the term $\phi_n = 0$. In this work, we will provide general results that depend on the value of $\phi_n$.

To facilitate the analysis, define the $\sigma$-filtration $\mathcal{F}_1 := \sigma(\theta_{1,1})$, $\mathcal{F}_n := \sigma(\sigma(\theta_{n-1,1}), \xi_{n-1})$ $(n \geq 2)$, where the random variable $\{\xi_n\}$ represents the stochasticity in the sampling of $\tilde{\mathbb{E}}$ and the estimation $\tilde{A}$ in step $n$ of the outer iterate.

### 3.2 ASSUMPTIONS

We first present the assumption regarding the policy parameterization of $\pi_\theta(s|a)$. This assumption can be realized by setting the policy to be within certain function classes such as neural networks.

**Assumption 3.1.** *There is a constant $L$ such that for any $s$, $a$, the policy $\pi_\theta(a|s)$ is $L$-smooth regarding $\theta$, i.e., $\|\nabla\pi_{\theta_1}(a|s) - \nabla\pi_{\theta_2}(a|s)\| \leq L\|\theta_1 - \theta_2\| \ \forall \theta_1, \theta_2$.*

In comparison, a stronger assumption was used in some previous works on policy gradient (Zhang et al., 2020; 2021; Yuan et al., 2020; Xu et al., 2020; Huang et al., 2020). In the assumption, the score function $\nabla \ln(\pi_\theta(s|a))$ has to be bounded and Lipschitz continuous, i.e., $\|\nabla \ln(\pi_{\theta_1}(s|a)) - \nabla \ln(\pi_{\theta_2}(s|a))\| \leq M_1\|\theta_1 - \theta_2\|$, for all $s, a$, for some $M_1$. This assumption does not hold for neural networks in general. As a counterexample, let the action space be $\{0, 1\}$ and let $\theta = (x, y)$. Let $\pi_{x,y}(0|s) = e^{-x^T ys}/(1 + e^{-x^T ys})$, $\pi_{x,y}(1|s) = 1/(1 + e^{-x^T ys})$, $(\forall s)$. When $\|\theta\| \to +\infty$, it is evident that $\|\nabla \ln(\pi_{\theta_1,\theta_2}(a|s))\| \to +\infty$ in this case. Nevertheless, our conditions in Assumption 3.1 hold under this example.

Our analyses also need a uniform upper bound for the reward function for each state and action. This condition is very commonly used in the field of reinforcement learning (Zhang et al., 2021; 2020; Yuan et al., 2020; Xu et al., 2020; Agarwal et al., 2021).

**Assumption 3.2.** *There exists an upper bound $\hat{r} > 0$, such that for any $s$, $a$, the reward is bounded by $|r(s,a)| \leq \hat{r}$.*

It follows immediately this assumption that $|A_\pi(s,a)| \leq R_{\max} := 2\hat{r}/(1-\gamma)$. In this manuscript, whenever $A$ is estimated (denoted as $\tilde{A}$), we truncate the value of that estimate such that $|\tilde{A}_\pi(s,a)| \leq R_{\max} = 2\hat{r}/(1-\gamma)$.

Now we describe the conditions of the learning rate of PPO-Clip. In this work, we require the learning rate to satisfy the Robbins-Monro condition, in order to prove the global convergence of PPO-Clip to a stationary point. The condition agrees with many commonly used learning rates of PPO in practice.

**Assumption 3.3.** *The learning rate* $\{\epsilon_{n,k}\}_{n=1,k=1}^{+\infty,T-1}$ *satisfies* $\sum_{n=1}^{+\infty} \sum_{k=1}^{T-1} \epsilon_{n,k} = +\infty$, $\sum_{n=1}^{+\infty} \sum_{k=1}^{T-1} \epsilon_{n,k}^2 < +\infty$.

In the case we only need to prove subsequence convergence or average-iterate mean-square Convergence, Assumption 3.3 can be relaxed into Assumption 3.4.

**Assumption 3.4.** *The learning rate* $\{\epsilon_{n,k}\}_{n=1,k=1}^{+\infty,T-1}$ *satisfies* $\sum_{n=1}^{+\infty} \sum_{k=1}^{T-1} \epsilon_{n,k} = +\infty$, $\sum_{k=1}^{T-1} \epsilon_{n,k} \to 0$.

Under Assumption 3.3, we can use the classical martingale method to prove convergence, which can yield strong last-iterate convergence. However, this approach is not applicable to the case under Assumption 3.4. Under Assumption 3.4, we are unable to construct a sup-martingale by the value of $V(\theta_n)$. Instead, we will need to decompose $\{\nabla V(\theta_n)\}$ into multiple sub-processes using first entrance times. Then a respective sup-martingale was constructed on each sub-process, and the compounding effect of these sub-processes is analyzed.

Assumptions on the learning rate can of course be satisfied by our choice of the learning rate. We remark that both the assumptions host a wide range of learning rate choices used in practice in reinforcement learning.

## 3.3 RESULTS

In investigating the convergence of stochastic optimization, it is common to construct a recursive inequality for the optimization objective. Taking policy gradient as an example, one may construct a recursive inequality using Taylor's expansion as

$$\mathbb{E}\left(V^* - V(\theta_{n+1})\big|\mathcal{F}_n\right) \leq V^* - V(\theta_n) - \epsilon_n \hat{\nabla} V(\theta_n)^T \nabla V(\theta_n) + \frac{\mathcal{L}\epsilon_n^2}{2}\|\hat{\nabla} V(\theta_n)\|^2,$$

where $\epsilon_n$ is the learning rate, $\hat{\nabla} V$ is a estimation of $\nabla V$, $V^*$ is the maximum value of the value function $V(\theta)$, and $\mathcal{L}$ represents the Lipschitz coefficient of $\nabla V(\theta)$.

Due to the difference between the policy gradient and PPO-Clip, the update rule now is not by a direct estimate of the policy gradient $\nabla V(\theta)$. In this case, to obtain a recursive inequality similar to the above one, we estimate the error between the PPO-Clip update target and $\nabla V(\theta)$. The aim is to bound this error term and mitigate its effect in the following analyses. However, it is not straightforward to estimate this error as one needs to estimate the distribution of the random variable $\pi_\theta(a|s)/\pi_{\theta_{old}}(a|s)$, which was not available in the literature to the best of our knowledge. From this perspective, the analysis of PPO-Clip might be more challenging compared to the analysis of policy gradient.

To be specific, our idea is to prove that the event $\overline{B} := \{\pi_\theta(a|s)/\pi_{\theta_{old}}(a|s) \in (0, 1-\delta_0) \cup (1+\delta_0, +\infty)\}$ is a subset of the event $\overline{C} := \{\sqrt{\pi_{\theta_{old}}(a|s)} \leq k_\theta\}$, for some scale $k_\theta$. Then we relax the set of $(s,a)$ pairs that satisfy the clip condition to those that satisfy the event $\overline{C}$. Subsequently, conditionally under the event $\overline{C}$ holds, we can estimate the error terms (denoted as $X$ and $Y$ in the analysis) accurately. This idea generates the following lemma.

**Lemma 3.1.** *If Assumption 3.1 and 3.2 hold, then for the sequence $\{\theta_{n,k}\}_{n=1,k=1}^{+\infty,T}$ formed by PPO-Clip (2), there is*

$$V^* - V(\theta_{n+1,1}) - (V^* - V(\theta_{n,1})) \leq -\left(\sum_{k=1}^{T-1} \epsilon_{n,k}\right)\|\nabla V(\theta_{n,1})\|^2$$

$$+ 4\sqrt{1+\delta_0}R_{max}^2\sqrt{|\mathcal{A}|}\mathcal{L}\left(\sum_{k=1}^{T-1}\epsilon_{n,k}\right)\left(\sum_{k=1}^{T-1}\epsilon_{n+1,k}\right)\|\nabla V(\theta_{n,1})\|^2$$

$$+ 2\sqrt{|\mathcal{A}|}\mathcal{L}\left(\phi_n + O\left(\sum_{k=1}^{T-1}\epsilon_{n,k}\right)\right)\cdot\left(\sum_{k=1}^{T-1}\epsilon_{n,k}\right) + \zeta_n,$$

*where*

$$\mathcal{L} := \frac{|\mathcal{A}|R_{max}L}{(1-\gamma)^2} + \frac{(1+\gamma)|\mathcal{A}|R_{max}\cdot\sqrt{2L}}{(1-\gamma)^3},$$

*and $\{\zeta_n, \mathcal{F}_{n+1}\}_{n=1}^{+\infty}$ is a martingale-difference sequence.*

This lemma is our key lemma. It derives a recursive inequity that is similar to that of policy gradient. We now provide the insights into finding this lemma, while deferring the full proof to the appendix.

**Step 1**: We first aim to simplify the gradient in PPO using a divide-and-conquer approach. Let

$$B_{n,k} := \left\{1 - \delta_0 < \frac{\pi_{\theta_{n,k}}(a|s)}{\pi_{\theta_{n,1}}(a|s)} < 1 + \delta_0\right\}$$

be the set of $(s, a)$ pairs that do not satisfy the clip operation. We obtain the following equations.

$$\nabla_{\theta_{n,k}}\left(\tilde{\mathbb{E}}_{\pi_{\theta_{n,1}}}\left(\min\left\{\text{clip}\left(\frac{\pi_{\theta_{n,k}}(a|s)}{\pi_{\theta_{n,1}}(a|s)}\right), \frac{\pi_{\theta_{n,k}}(a|s)}{\pi_{\theta_{n,1}}(a|s)}\right\}\cdot\tilde{A}_{\pi_{\theta_{n,1}}}(s,a)\right)\right)$$

$$= \tilde{\mathbb{E}}_{\pi_{\theta_{n,1}}}\left(\mathbf{1}_{B_{n,k}^{(s,a)}}\frac{\nabla\pi_{\theta_{n,k}}(a|s)}{\pi_{\theta_{n,1}}(a|s)}\cdot\tilde{A}_{\pi_{\theta_{n,1}}}(s,a)\right)$$

$$= \tilde{\mathbb{E}}_{\pi_{\theta_{n,1}}}\left(\frac{\nabla\pi_{\theta_{n,1}}(a|s)}{\pi_{\theta_{n,1}}(a|s)}\cdot\tilde{A}_{\pi_{\theta_{n,1}}}(s,a)\right) - \tilde{\mathbb{E}}_{\pi_{\theta_{n-1}}}\left(\mathbf{1}_{\Omega/B_{n,k}^{(s,a)}}\frac{\nabla\pi_{\theta_{n,1}}(a|s)}{\pi_{\theta_{n,1}}(a|s)}\cdot\tilde{A}_{\pi_{\theta_{n,1}}}(s,a)\right)$$

$$+ \tilde{\mathbb{E}}_{\pi_{\theta_{n,1}}}\left(\mathbf{1}_{B_{n,k}^{(s,a)}}\frac{\nabla\pi_{\theta_{n,k}}(a|s) - \nabla\pi_{\theta_{n,1}}(a|s)}{\pi_{\theta_{n,1}}(a|s)}\cdot\tilde{A}_{\pi_{\theta_{n,1}}}(s,a)\right)$$

$$:= \tilde{\mathbb{E}}_{\pi_{\theta_{n,1}}}\left(\frac{\nabla\pi_{\theta_{n,1}}(a|s)}{\pi_{\theta_{n,1}}(a|s)}\cdot\tilde{A}_{\pi_{\theta_{n,1}}}(s,a)\right) + X_{n,k} + Y_{n,k}.$$

Here, $X_{n,k}$ and $Y_{n,k}$ are defined as the last two terms, i.e. the error terms, on the right-hand side of the second equation.

**Step 2**: Note that the first term of the last equation is the policy gradient of the value function at $\theta_{n,1}$, which can be handled effectively. Therefore, we aim to bound $X_{n,k}$ and $Y_{n,k}$ to ensure that the difference between the gradient of PPO and the policy gradient of the value function at $\theta_{n,1}$ is not significant.

For $X_{n,k}$, we observe $\Omega/B_{n,k} \subset \Omega/C_{n,k}$ (see appendix for its proof), where

$$C_{n,k} := \left\{\sqrt{\pi_{\theta_{n,1}}(a|s)} > (\sqrt{2L}+\lambda_0)\cdot\left(\sum_{i=1}^{k-1}\epsilon_{n-1,i}\right)\cdot\tilde{\mathbb{E}}_{\pi_{\theta_{n-1,1}}}\left(\frac{\sqrt{2L(1+\delta_0)}}{\sqrt{\pi_{\theta_{n-1,1}}(a|s)}}\cdot\left|\tilde{A}_{\pi_{\theta_{n-1,1}}}(s,a)\right|\right)\right\}$$

and $\lambda_0 := \min\left\{\frac{\sqrt{2L}}{-\ln(1-\delta_0)}, \frac{\sqrt{2L}}{\ln(1+\delta_0)}\right\}$. As such, the indicator function $\mathbf{1}_{B_{n,k}}$ can be bounded by $\mathbf{1}_{C_{n,k}}$. By further noticing the $L$-smooth condition, we have $\|\nabla\pi_{\theta_{n,k}}(a|s)\| \leq \sqrt{2L\pi_{\theta_{n,k}}(a|s)}$ (see

Lemma A.1 for a proof), which leads to

$$\mathbb{E}\left(\|X_{n,k}\|\big|\mathcal{F}_{n-1}\right) \leq 2L\sqrt{1+\delta_0}R_{\max}^3|\mathcal{A}|\left(\sum_{i=1}^{k-1}\epsilon_{n,i}\right)\tilde{\mathbb{E}}_{\pi_{\theta_{n-1,1}}}\left(\frac{1}{\sqrt{\pi_{\theta_{n-1,1}}(a|s)}}\right) + \sqrt{|\mathcal{A}|}R_{\max}\phi_n,$$

where $R_{\max} := \hat{r}/(1-\gamma)$ is defined as an upper bound on the value function.

For $Y_n$, our main objective is to bound $\nabla\pi_{\theta_{n,k}}(a|s) - \nabla\pi_{\theta_{n,1}}(a|s)$. According to the $L$-smooth condition, we have $\|\nabla\pi_{\theta_{n,k}}(a|s) - \nabla\pi_{\theta_{n,1}}(a|s)\| \leq L\|\theta_{n,k} - \theta_{n,1}\|$. It therefore suffices to control $\|\theta_{n,k} - \theta_{n,1}\|$. In fact, we could obtain

$$\|\theta_{n,k+1} - \theta_{n,k}\| \leq \epsilon_{n,k}\tilde{\mathbb{E}}_{\pi_{\theta_{n,1}}}\left(\frac{\sqrt{2L(1+\delta_0)}}{\sqrt{\pi_{\theta_{n,1}}(a|s)}}\cdot\left|\tilde{A}_{\pi_{\theta_{n,1}}}(s,a)\right|\right),$$

and the proof of this inequality is presented in Lemma A.2. The bound of $Y_n$ is then

$$\mathbb{E}\left(\|Y_n\|\big|\mathcal{F}_{n-1}\right) \leq L\sqrt{2L(1+\delta_0)}R_{\max}^3|\mathcal{A}|\left(\sum_{i=1}^{k-1}\epsilon_{n,i}\right)\tilde{\mathbb{E}}_{\pi_{\theta_{n-1,1}}}\left(\frac{1}{\sqrt{\pi_{\theta_{n-1,1}}(a|s)}}\right) + \sqrt{|\mathcal{A}|}R_{\max}\phi_n.$$

**Step 3**: In this step, we construct a recursive equation for the value function $V(\theta_{n,1})$. Using Taylor's expansion, we obtain

$$
\begin{aligned}
&V^* - V(\theta_{n+1,1}) - (V^* - V(\theta_{n,1})) \\
&\leq \nabla V(\theta_{n-1})^\top(\theta_{n+1,1} - \theta_{n,1}) + \left(\nabla V(\theta_{n,1}) - \nabla V(\theta_{n-1})\right)^\top(\theta_{n+1,1} - \theta_{n,1}) \\
&\quad + \mathcal{L}\|\theta_{n+1,1} - \theta_{n,1}\|^2,
\end{aligned}
$$

where $\mathcal{L} := \frac{|\mathcal{A}|R_{\max}L}{(1-\gamma)^2} + \frac{(1+\gamma)|\mathcal{A}|R_{\max}\cdot\sqrt{2L}}{(1-\gamma)^3}$ represents the Lipschitz coefficient of $\nabla V(\theta)$. By substituting the expressions for $X_{n,k}$ and $Y_{n,k}$ from the previous step into the equation, there is

$$
\begin{aligned}
V^* - V(\theta_{n+1,1}) - (V^* - V(\theta_{n,1})) &\leq -\left(\sum_{k=1}^{T-1}\epsilon_{n,k}\right)\|\nabla V(\theta_{n,1})\|^2 \\
&+ 4\sqrt{1+\delta_0}R_{\max}^2\sqrt{|\mathcal{A}|}\mathcal{L}\left(\sum_{k=1}^{T-1}\epsilon_{n,k}\right)\left(\sum_{k=1}^{T-1}\epsilon_{n+1,k}\right)\|\nabla V(\theta_{n,1})\|^2 \\
&+ 2\sqrt{|\mathcal{A}|}\mathcal{L}\left(\phi_n + O\left(\sum_{k=1}^{T-1}\epsilon_{n,k}\right)\right)\cdot\left(\sum_{k=1}^{T-1}\epsilon_{n,k}\right).
\end{aligned}
$$

Armed with the lemma, we are now ready to discuss the convergence of the PPO-Clip algorithm. Our first theorem asserts its global convergence to a stationary point, up to the estimation bias $\phi_n$ involved in the Markovian sampling and the advantage estimation.

The idea of the proof is to discuss the magnitude of $\|\nabla V(\theta_{n,1})\|^2$. Let $\phi_n' := \phi_n + O\left(\sum_{k=1}^{T-1}\epsilon_{n,k}\right)$. When $\|\nabla V(\theta_{n,1})\|^2 > 8\sqrt{|\mathcal{A}|}\phi_n'$, the fixed bias term $2\sqrt{|\mathcal{A}|}\mathcal{L}\phi_n'\left(\sum_{k=1}^{T-1}\epsilon_{n,k}\right)$ can be immediately offset by $\left(\sum_{k=1}^{T-1}\epsilon_{n,k}\right)\|\nabla V(\theta_{n-1})\|^2$. When $\|\nabla V(\theta_{n,1})\|^2 \leq 8\sqrt{|\mathcal{A}|}\mathcal{L}\phi_n'$, the desired result is straightforwardly satisfied.

**Theorem 3.1.** *If Assumption 3.1, 3.2, 3.4 hold, then for the sequence $\{\theta_{n,k}\}_{n=1,k=1}^{+\infty,T}$ formed by PPO-Clip (2), there is*

$$\liminf_{n\to+\infty}\|\nabla V(\theta_{n,1})\|^2 \leq 8\mathcal{L}\sqrt{|\mathcal{A}|}\limsup_{n\to+\infty}\phi_n \quad a.s..$$

*Proof.* Let $\phi'_n := \phi_n + O\big(\sum_{k=1}^{T-1} \epsilon_{n,k}\big)$. We construct the following events.

$$\mathcal{A}_{i,n} := \big\{\{\|\nabla V(\theta_{i,1})\|^2 > \max\{8\sqrt{|\mathcal{A}|}\sup_{t\geq i} \phi'_t, \delta\}\}, \{\|\nabla V(\theta_{i+1,1})\|^2 > 8\sqrt{|\mathcal{A}|}\sup_{t\geq i+1} \phi'_t \phi'_{i+1}\},$$

$$..., \{\|\nabla V(\theta_{n,1})\|^2 > \max\{8\sqrt{|\mathcal{A}|}\sup_{t\geq i} \phi'_t, \delta\}\},$$

$$\mathcal{A}_{i,i} := \{\|\nabla V(\theta_{i,1})\|^2 > \max\{8\sqrt{|\mathcal{A}|}\sup_{t\geq i} \phi'_t, \delta\}\},$$

$$\mathcal{A}_{i,+\infty} := \big\{\{\|\nabla V(\theta_{i,1})\|^2 > \max\{8\sqrt{|\mathcal{A}|}\sup_{t\geq i} \phi'_t, \delta\}\}, \{\|\nabla V(\theta_{i+1,1})\|^2 > 8\sqrt{|\mathcal{A}|}\sup_{t\geq i+1} \phi'_t\},$$

$$..., \{\|\nabla V(\theta_{n,1})\|^2 > \max\{8\sqrt{|\mathcal{A}|}\sup_{t\geq n} \phi'_t, \delta\}, ...\}.$$

where $\hat{V} := V^* - V$ and $\delta$ is a positive constant. Asymptotically, we can obtain the existence of a $N_0$ such that for $n > N_0$, there is

$$\Big(\sum_{k=1}^{T-1} \epsilon_{n+1,k}\Big) \mathbb{E}\,\|\nabla V(\theta_{n,1})\|^2 > 8\sqrt{1+\delta}R_{\max}^2\sqrt{|\mathcal{A}|}\mathcal{L}\Big(\sum_{k=1}^{T-1} \epsilon_{n,k}\Big)\Big(\sum_{k=1}^{T-1} \epsilon_{n+1,k}\Big)\mathbb{E}\,\|\nabla V(\theta_{n,1})\|^2.$$

Note that here $N_0$ only depends on the parameters $\delta$, $R_{\max}$, $|\mathcal{A}|$, $\mathcal{L}$ and the learning rate $\epsilon_{n,k}$ itself. Then through Equation (14) in the proof of Lemma 3.1, when $\forall\, i > T_0$, $n \geq i$, $\mathcal{X} \in \mathcal{F}_i$, there is

$$\mathbf{1}_{\mathcal{X}\cap\mathcal{A}_{i,n}}\hat{V}(\theta_{n+1,1}) - \mathbf{1}_{\mathcal{X}\cap\mathcal{A}_{i,n}}\hat{V}(\theta_{n,1}) \leq -\frac{1}{4}\Big(\sum_{k=1}^{T-1} \epsilon_{n,k}\Big)\cdot\mathbf{1}_{\mathcal{X}\cap\mathcal{A}_{i,n}}\|\nabla V(\theta_{n,1})\|^2 + \mathbf{1}_{\mathcal{X}\cap\mathcal{A}_{i,n}}\zeta_n.$$

Notice $\mathbf{1}_{\mathcal{X}\cap\mathcal{A}_{i,n}}\hat{V}(\theta_{n+1,1}) \geq \mathbf{1}_{\mathcal{X}\cap\mathcal{A}_{i,n+1}}\hat{V}(\theta_{n+1,1})$. Then,

$$\mathbf{1}_{\mathcal{X}\cap\mathcal{A}_{i,n+1}}\hat{V}(\theta_{n+1,1}) - \mathbf{1}_{\mathcal{X}\cap\mathcal{A}_{i,n}}\hat{V}(\theta_{n,1}) \leq -\frac{1}{4}\Big(\sum_{k=1}^{T-1} \epsilon_{n,k}\Big)\cdot\mathbf{1}_{\mathcal{X}\cap\mathcal{A}_{i,n}}\|\nabla V(\theta_{n,1})\|^2 + \mathbf{1}_{\mathcal{X}\cap\mathcal{A}_{i,n}}\zeta_n.$$

By summing up the above expression from $i$ to $t$, we obtain

$$\frac{\delta}{4}\sum_{n=i}^{t}\Big(\sum_{k=1}^{T-1} \epsilon_{n,k}\Big)\mathbb{E}(\mathbf{1}_{\mathcal{X}\cap\mathcal{A}_{i,n}}) \leq +\infty.$$

Combining the condition $\sum_{n=i}^{t}\big(\sum_{k=1}^{T-1} \epsilon_{n,k}\big) = +\infty$, and the fact the sequence $\{\mathbb{E}(\mathbf{1}_{\mathcal{X}\cap\mathcal{A}_{i,n}})\}_{n=i}^{+\infty}$ has a limit, we have

$$\lim_{n\to+\infty}\mathbb{E}(\mathbf{1}_{\mathcal{X}\cap\mathcal{A}_{i,n}}) = 0,$$

which implies $\mathbb{E}(\mathbf{1}_{\mathcal{X}\cap\mathcal{A}_{i,+\infty}}) = 0$. According to the arbitrariness of $\delta$, $i$ and $\mathcal{X}$, we get $\liminf_{n\to+\infty}\|\nabla V(\theta_{n,1})\|^2 \leq 8\mathcal{L}\sqrt{|\mathcal{A}|}\limsup_{n\to+\infty}\phi_n$ $a.s.$. $\qquad\square$

This theorem essentially provides us the guarantee that PPO-Clip makes the gradient norm of the value function subsequence converge almost surely to zero within an $O(\limsup_{n\to+\infty}\phi_n)$ neighborhood. In practice, the size of this neighborhood is determined by the advantage network. The more accurate is the advantage function estimation, the smaller is this neighborhood. When the advantage function is estimated unbiasedly, like through Monte-Carlo sampling, the convergence is exact.

In fact, when $\phi_n = 0$, we further improve this result by replacing Assumption 3.4 with Assumption 3.3, which is strictly weaker.

**Theorem 3.2.** *If Assumption 3.1, 3.2, 3.3 hold and there exists a $t > 0$, such that $\phi_n = 0$ $(\forall\, n > t)$, then for the sequence $\{\theta_{n,k}\}_{n=1,k=1}^{+\infty,T}$ formed by PPO-Clip (2), there are*

$$\lim_{n\to+\infty}\|\nabla V(\theta_n)\| = 0 \ a.s., \ and \ \lim_{n\to+\infty}\mathbb{E}\,\|\nabla V(\theta_n)\|^2 = 0.$$

This theorem indicates that PPO-Clip has the same convergence property as the best current results available for policy gradient (Zhang et al., 2020). The proof of this theorem is by Lemma 3.1 and classic martingale methods in stochastic approximation (Robbins & Monro, 1951). We defer the proof to Appendix A.3.

Now we proceed to present the results regarding the convergence rate. Unlike existing research on the convergence of the policy gradient algorithm which acquired the last-iterate convergence rate (Zhang et al., 2020; 2021; Yuan et al., 2020; Xu et al., 2020; Huang et al., 2020), i.e., $\mathbb{E} \|\nabla V(\theta_n)\|^2 = O(\cdot)$, we provide the convergence rate in terms of the average iterate, i.e., $1/(\sum_{k=1}^T \epsilon_k) \sum_{k=1}^T \epsilon_k \mathbb{E} \|\nabla V(\theta_k)\|^2 = O(\cdot)$.

We remark that this difference is due to the conditions of the analysis. In general, for stochastic gradient descent algorithms, in order to provide convergence rate in terms of the last-iterate sense, it is necessary to establish a local or global inequality that connects $\nabla V$ and $V$. In the study of SGD-type algorithms, examples of such inequalities are P-L conditions and K-L conditions. In the study of the PG algorithm, assumptions like *bounded score functions*, *Fisher non-degenerate assumption*, and *transfer error bounded conditions* are often used, which then imply the gradient dominance condition, i.e., $\|\nabla V(\theta)\| \geq V^* - V(\theta) + \epsilon_{\text{bias}}$. This condition is similar to the K-L condition. As we have not imposed any such additional conditions, the convergence rate in terms of the average-iterate convergence is the best we could have obtained.

**Theorem 3.3.** *If Assumption 3.1, 3.2, 3.4 hold, then for the sequence $\{\theta_{n,k}\}_{n=1,k=1}^{+\infty,T}$ formed by PPO (2), there is*

$$\min_{m=1,2,\ldots,n} \left\{ \mathbb{E} \|\nabla V(\theta_{t,1})\|^2 \right\} = O\left( \frac{\sum_{t=1}^n \tilde{\epsilon}_t^2}{\sum_{t=1}^n \tilde{\epsilon}_t} \right) + O\left( \limsup_{n \to +\infty} \phi_n \right),$$

*and*

$$\frac{1}{\sum_{t=1}^n \tilde{\epsilon}_t} \sum_{t=1}^n \tilde{\epsilon}_t \mathbb{E} \|\nabla V(\theta_{t,1})\|^2 = O\left( \frac{\sum_{t=1}^n \tilde{\epsilon}_t^2}{\sum_{t=1}^n \tilde{\epsilon}_t} \right) + O\left( \limsup_{n \to +\infty} \phi_n \right),$$

*where $\tilde{\epsilon}_t := \sum_{k=1}^{T-1} \epsilon_{t,k}$.*

*Proof.* By Lemma 3.1, we have

$$\mathbb{E}(V^* - V(\theta_{n+1,1})) - \mathbb{E}(V^* - V(\theta_{n,1})) \leq -\left( \sum_{k=1}^{T-1} \epsilon_{n,k} \right) \mathbb{E} \|\nabla V(\theta_{n,1})\|^2$$

$$+ 4\sqrt{1+\delta_0} R_{\max}^2 \sqrt{|\mathcal{A}|} \mathcal{L} \left( \sum_{k=1}^{T-1} \epsilon_{n,k} \right)^2 \mathbb{E} \|\nabla V(\theta_{n,1})\|^2$$

$$+ 2\sqrt{|\mathcal{A}|} \mathcal{L} \left( \sum_{k=1}^{T-1} \epsilon_{n,k} \right) \phi_n + O\left( \left( \sum_{k=1}^{T-1} \epsilon_{n,k} \right)^2 \right),$$

where $\tilde{\epsilon}_t := \sum_{k=1}^{T-1} \epsilon_{t,k}$. Then, we obtain

$$\min_{m=1,2,\ldots,n} \left\{ \mathbb{E} \|\nabla V(\theta_{t,1})\|^2 \right\} \leq \frac{1}{\sum_{t=1}^n \tilde{\epsilon}_t} \sum_{t=1}^n \tilde{\epsilon}_t \mathbb{E} \|\nabla V(\theta_{t,1})\|^2 \leq O\left( \frac{\sum_{t=1}^n \tilde{\epsilon}_t^2}{\sum_{t=1}^n \tilde{\epsilon}_t} \right) + O\left( \limsup_{n \to +\infty} \phi_n \right).$$

The convergence rate under the notion of best-iterate convergence and average-iterate convergence follow, respectively. □

Our analyses might provide some insights that PPO could be more efficient than PG, in some scenarios. To compare them, we set the learning rate in PPO-Clip as $\epsilon_{n,k} = \epsilon_n$ ($\forall \, 1 \leq k \leq T$), and the learning rate in PG as $\epsilon_n$. The convergence rate of PPO-Clip is $O\left( \frac{T \sum_{t=1}^n \epsilon_n^2}{\sum_{t=1}^n \epsilon_t} \right) + O(\limsup_{n \to +\infty} \phi_n)$,, while the convergence rate of PG is $O\left( \frac{\sum_{t=1}^n \epsilon_n^2}{\sum_{t=1}^n \epsilon_t} \right) + O(\limsup_{n \to +\infty} \phi_n)$. One could observe that both the convergence rates are in the same order, up to a factor of $T$, which is the number of steps of off policy updates using $\pi_{\text{old}}$. In this way, while PG requires continuous sampling to obtain on policy data, PPO-Clip use one sample for $T$ steps. Improving the sample efficiency is critical in many reinforcement learning tasks, and when sampling is the bottleneck, PPO-Clip could have an edge over PG.

# 4 CONCLUSION AND FUTURE WORKS

We investigate PPO-Clip, one of the most popular RL algorithms yet to be fully understood in theory. We show that PPO-Clip converges to stationary points up to a bias term, and when the biases in Markovian sampling and in advantage estimation are zero this bias term is also zero. We further proved the convergence rate of average-iterate convergence and subsequence convergence.

We used a minimum set of conditions in our analysis. The conditions on step sizes and policy parametrization could be satisfied by the choices of step sizes and policy class. The condition on bounded reward function is rather common and mild in RL analysis. The relaxation of conditions incurs several technical challenges to overcome, which motivates us to introduce our key lemma, Lemma 3.1. Our lemma derives the recursive property of the optimization process under the presence of the clipped ratio of two stochastic policies, and could be of independent interest for analysis on other optimization algorithms.

It remains unknown to us if the optimality of PPO-Clip could be deduced under the current conditions, but it would be important for future works to further characterize the optimality of the stationary points that PPO-Clip converges to. Another direction is to investigate other conditions on the policy, such that it is general enough to include neural networks but provides improved results on the convergence and the optimality.

## ACKNOWLEDGEMENT

Ruinan Jin and Baoxiang Wang are partially supported by National Natural Science Foundation of China (62106213, 72150002, 72394361) and Shenzhen Science and Technology Program (RCBS20210609104356063, JCYJ20210324120011032). The work is partially supported by Guangdong Provincial Key Laboratory of Big Data Computing of The Chinese University of Hong Kong, Shenzhen.

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

# A APPENDIX

## A.1 AUXILIARY LEMMAS

**Lemma A.1.** *If Assumption 3.1 holds, we have*

$$\left\|\nabla\pi_\theta(a|s)\right\| \leq \sqrt{2L \cdot \min\{\pi_\theta(a|s),\ 1 - \pi_\theta(a|s)\}}\ \ (\forall\ s,\ a).$$

*Proof.* By applying Lemma 10 of Jin et al. (2022), we get

$$\left\|\nabla\pi_\theta(a|s)\right\| \leq \sqrt{2L(\pi_\theta(a|s) - \inf_{\theta'\in\mathbb{R}^d}\pi_{\theta'}(a|s))} \leq \sqrt{2L\pi_\theta(a|s)}. \tag{4}$$

Then we construct a function $f^{(s,a)}(\theta) := 1 - \pi_\theta(a|s)$. By its definition we have $\nabla f^{(s,a)}(\theta) = -\nabla\pi_\theta(a|s)$. By Assumption 3.1, we have for any $\theta_1,\ \theta_2$, such that

$$\left\|\nabla f^{(s,a)}(\theta_1) - \nabla f^{(s,a)}(\theta_2)\right\| = \left\|\pi_{\theta_1}(a|s) - \pi_{\theta_2}(a|s)\right\| \leq L\|\theta_1 - \theta_2\|.$$

Similarly, we obtain

$$\left\|\nabla f^{(s,a)}(\theta)\right\| \leq \sqrt{2Lf^{(s,a)}(\theta) - \inf_{\theta'\in\mathbb{R}^d} f^{(s,a)}(\theta')} \leq \sqrt{2Lf^{(s,a)}(\theta)},$$

that is

$$\left\|\pi_\theta(a|s)\right\| \leq \sqrt{2L(1 - \pi_\theta(a|s))}. \tag{5}$$

Combining Equation (4) and Equation (5) yields the desired inequality. $\square$

**Lemma A.2.** *If Assumption 3.1 holds, then for the sequence $\{\theta_{n,k}\}_{n=1,k=1}^{+\infty,T}$ formed by PPO (2), there is $\forall\ n > 0,\ \forall\ 0 < k < T$,*

$$\|\theta_{n,k+1} - \theta_{n,k}\| \leq \epsilon_{n,k}\tilde{\mathbb{E}}_{\pi_{\theta_{n,1}}}\left(\frac{\sqrt{2L(1+\delta_0)}}{\sqrt{\pi_{\theta_{n,1}}(a|s)}} \cdot \left|\tilde{A}_{\pi_{\theta_{n-1}}}(s,a)\right|\right).$$

*Proof.* By the update process of PPO in Equation (2), we have

$$\|\theta_{n,k+1} - \theta_{n,k}\| = \epsilon_{n,k}\left\|\nabla_{\theta_{n,k}}\left(\tilde{\mathbb{E}}_{\pi_{\theta_{n,1}}}\left(\text{clip}\left(\frac{\pi_{\theta_{n,k}}(a|s)}{\pi_{\theta_{n,1}}(a|s)}\right) \cdot \tilde{A}_{\pi_{\theta_{n,1}}}(s,a)\right)\right)\right\|.$$

Define events

$$B_{n,k} := \left\{1 - \delta_0 < \frac{\pi_{\theta_{n,k}}(a|s)}{\pi_{\theta_{n,1}}(a|s)} < 1 + \delta_0\right\},$$

and its indicator function as $\mathbf{1}_{B_{n,k}^{(s,a)}}$. Then,

$$\begin{aligned}
\epsilon_{n,k}&\left\|\tilde{\mathbb{E}}_{\pi_{\theta_{n,1}}}\left(\mathbf{1}_{B_{n,k}^{(s,a)}}\frac{\nabla\pi_{\theta_{n,1}}(a|s)}{\pi_{\theta_{n,1}}(a|s)} \cdot \tilde{A}_{\pi_{\theta_{n,1}}}(s,a)\right)\right\| \\
&\leq \epsilon_{n,k}\tilde{\mathbb{E}}_{\pi_{\theta_{n,1}}}\left(\mathbf{1}_{B_{n,k}^{(s,a)}}\frac{\|\nabla\pi_{\theta_{n,1}}(a|s)\|}{\pi_{\theta_{n,1}}(a|s)} \cdot \left|\tilde{A}_{\pi_{\theta_{n,1}}}(s,a)\right|\right).
\end{aligned} \tag{6}$$

By Lemma A.1,

$$\|\nabla\pi_{\theta_{n,1}}(a|s)\| \leq \sqrt{2L\pi_{\theta_{n,1}}(a|s)}.$$

This implies

$$\|\theta_{n,k+1} - \theta_{n,k}\| \leq \epsilon_{n,k}\tilde{\mathbb{E}}_{\pi_{\theta_{n,1}}}\left(\frac{\sqrt{2L(1+\delta_0)}}{\sqrt{\pi_{\theta_{n,1}}(a|s)}} \cdot \left|\tilde{A}_{\pi_{\theta_{n,1}}}(s,a)\right|\right).$$

The lemma follows. $\square$

## A.2 PROOF OF LEMMA 3.1

*Proof.* We continue to use the events $B_{n,k}^{(s,a)}$ and the indicator function $\mathbf{1}_{B_{n,k}^{(s,a)}}$ defined in the proof of Lemma A.2. Using these notations, we rewrite the gradient term in PPO as

$$
\nabla_{\theta_{n,k}}\left(\tilde{\mathbb{E}}_{\pi_{\theta_{n,1}}}\left(\text{clip}\left(\frac{\pi_{\theta_{n,k}}(a|s)}{\pi_{\theta_{n,1}}(a|s)}\right)\cdot\tilde{A}_{\pi_{\theta_{n,1}}}(s,a)\right)\right)=\tilde{\mathbb{E}}_{\pi_{\theta_{n,1}}}\left(\mathbf{1}_{B_{n,k}^{(s,a)}}\frac{\nabla\pi_{\theta_{n,k}}(a|s)}{\pi_{\theta_{n,1}}(a|s)}\cdot\tilde{A}_{\pi_{\theta_{n,1}}}(s,a)\right)
$$

$$
=\tilde{\mathbb{E}}_{\pi_{\theta_{n,1}}}\left(\frac{\nabla\pi_{\theta_{n,1}}(a|s)}{\pi_{\theta_{n,1}}(a|s)}\cdot\tilde{A}_{\pi_{\theta_{n,1}}}(s,a)\right)-\tilde{\mathbb{E}}_{\pi_{\theta_{n-1}}}\left(\mathbf{1}_{\Omega/B_{n,k}^{(s,a)}}\frac{\nabla\pi_{\theta_{n,1}}(a|s)}{\pi_{\theta_{n,1}}(a|s)}\cdot\tilde{A}_{\pi_{\theta_{n,1}}}(s,a)\right)
$$

$$
+\tilde{\mathbb{E}}_{\pi_{\theta_{n,1}}}\left(\mathbf{1}_{B_{n,k}^{(s,a)}}\frac{\nabla\pi_{\theta_{n,k}}(a|s)-\nabla\pi_{\theta_{n,1}}(a|s)}{\pi_{\theta_{n,1}}(a|s)}\cdot\tilde{A}_{\pi_{\theta_{n,1}}}(s,a)\right)
$$

$$
:=\tilde{\mathbb{E}}_{\pi_{\theta_{n,1}}}\left(\frac{\nabla\pi_{\theta_{n,1}}(a|s)}{\pi_{\theta_{n,1}}(a|s)}\cdot\tilde{A}_{\pi_{\theta_{n,1}}}(s,a)\right)+X_{n,k}+Y_{n,k}.
$$

$$(7)$$

Inspecting the norm of the term $X_n$ in Equation (7) yields

$$
\|X_{n,K}\|=\left\|-\tilde{\mathbb{E}}_{\pi_{\theta_{n,1}}}\left(\mathbf{1}_{\Omega/B_{n,k}^{(s,a)}}\frac{\nabla\pi_{\theta_{n,1}}(a|s)}{\pi_{\theta_{n,1}}(a|s)}\cdot\tilde{A}_{\pi_{\theta_{n,1}}}(s,a)\right)\right\|
$$

$$
\leq\tilde{\mathbb{E}}_{\pi_{\theta_{n,1}}}\left(\mathbf{1}_{\Omega/B_{n,k}^{(s,a)}}\frac{\|\nabla\pi_{\theta_{n,1}}(a|s)\|}{\pi_{\theta_{n,1}}(a|s)}\cdot\left|\tilde{A}_{\pi_{\theta_{n,1}}}(s,a)\right|\right).
$$

$$(8)$$

Now we aim to estimate the indicator $\mathbf{1}_{\Omega/B_{n,k}^{(s,a)}}$. Through the mean value theorem and Lemma A.1, we have

$$
\frac{\pi_{\theta_{n,k}}(a|s)}{\pi_{\theta_{n,1}}(a|s)}=e^{\ln\pi_{\theta_{n,k}}(a|s)-\ln\pi_{\theta_{n,1}}(a|s)}=\exp\left\{\left(\frac{\nabla_{\tilde{\theta}_{n,k}}\pi_{\tilde{\theta}_{n,k}}(a|s)}{\pi_{\tilde{\theta}_{n,k}}(a|s)}\right)^{\top}(\theta_{n,k}-\theta_{n,1})\right\}
$$

$$
\in\left(\exp\left\{-\frac{\|\nabla_{\tilde{\theta}_{n,k}}\pi_{\tilde{\theta}_{n,k}}(a|s)\|}{\pi_{\tilde{\theta}_{n,k}}(a|s)}\|\theta_{n,k}-\theta_{n,1}\|\right\},\ \exp\left\{\frac{\|\nabla_{\tilde{\theta}_{n,k}}\pi_{\tilde{\theta}_{n,k}}(a|s)\|}{\pi_{\tilde{\theta}_{n,k}}(a|s)}\|\theta_{n,k}-\theta_{n,1}\|\right\}\right)
$$

$$
\subset\left(\exp\left\{-\frac{\sqrt{2L}}{\sqrt{\pi_{\tilde{\theta}_{n,k}}(a|s)}}\cdot\|\theta_{n,k}-\theta_{n,1}\|\right\},\ \exp\left\{\frac{\sqrt{2L}}{\sqrt{\pi_{\tilde{\theta}_{n,k}}(a|s)}}\cdot\|\theta_{n,k}-\theta_{n,1}\|\right\}\right),
$$

$$(9)$$

where $\tilde{\theta}_{n,k}$ is a point between $\theta_{n,k}$ and $\theta_{n,1}$. We construct another event

$$
C_{n,k,\lambda}^{(s,a)}:=\left\{\sqrt{\pi_{\theta_{n,1}}(a|s)}>(\sqrt{2L}+\lambda)\cdot\left(\sum_{i=1}^{k-1}\epsilon_{n-1,i}\right)\cdot\tilde{\mathbb{E}}_{\pi_{\theta_{n-1,1}}}\left(\frac{\sqrt{2L(1+\delta_0)}}{\sqrt{\pi_{\theta_{n-1,1}}(a|s)}}\cdot\left|\tilde{A}_{\pi_{\theta_{n-1,1}}}(s,a)\right|\right)\right\},
$$

where $\lambda>0$ is a coefficient to be determined. It is worth noting that the event $C_{n,k,\lambda}^{(s,a)}$ belongs to the $\sigma-$field $\mathcal{F}_{n-1}$, which is used in the subsequent proof. Then,

$$
\sqrt{\pi_{\tilde{\theta}_{n,k}}(a|s)}=\sqrt{\pi_{\theta_{n,1}}(a|s)}+\sqrt{\pi_{\tilde{\theta}_{n,k}}(a|s)}-\sqrt{\pi_{\theta_{n,1}}(a|s)}
$$

$$
\geq\sqrt{\pi_{\theta_{n,1}}(a|s)}-\left|\sqrt{\pi_{\tilde{\theta}_{n,k}}(a|s)}-\sqrt{\pi_{\theta_{n,1}}(a|s)}\right|.
$$

By inspecting the function $\sqrt{\pi_\theta(a|s)}$, we obtain

$$
\left\|\nabla\sqrt{\pi_\theta(a|s)}\right\|=\frac{\|\nabla\pi_\theta(a|s)\|}{\sqrt{\pi_\theta(a|s)}}\leq\sqrt{2L}.
$$

That means

$$
\left|\sqrt{\pi_{\tilde{\theta}_{n,k}}(a|s)}-\sqrt{\pi_{\theta_{n,1}}(a|s)}\right|\leq\sqrt{2L}\cdot\|\tilde{\theta}_{n,k}-\theta_{n,1}\|\leq\sqrt{2L}\cdot\|\theta_{n,k}-\theta_{n,1}\|.
$$

Therefore, whenever $C_{n,k,\lambda}^{(s,a)}$ happens, through Lemma A.2, there is

$$
\begin{aligned}
\sqrt{\pi_{\tilde{\theta}_{n,k}}(a|s)} &\geq \sqrt{\pi_{\theta_{n,1}}(a|s)} - \sqrt{2L} \cdot \|\tilde{\theta}_{n,k} - \theta_{n,1}\| \\
&> (\sqrt{2L} + \lambda) \cdot \left( \sum_{i=1}^{k-1} \epsilon_{n-1,i} \right) \cdot \tilde{\mathbb{E}}_{\pi_{\theta_{n-1,1}}} \left( \frac{\sqrt{2L(1+\delta_0)}}{\sqrt{\pi_{\theta_{n-1,1}}(a|s)}} \cdot \left| \tilde{A}_{\pi_{\theta_{n-1,1}}}(s,a) \right| \right) - \sqrt{2L} \cdot \|\theta_{n,1} - \theta_{n-1}\| \\
&\geq (\sqrt{2L} + \lambda) \cdot \|\theta_{n,k} - \theta_{n,1}\| - \sqrt{2L} \cdot \|\theta_{n,1} - \theta_{n-1}\| \\
&= \lambda \cdot \|\theta_{n,k} - \theta_{n,1}\|.
\end{aligned}
$$

Substituting the above inequity into Equation (9), we have that whenever $C_{n,k,\lambda}^{(s,a)}$ happens, there is

$$
\frac{\pi_{\theta_{n,k}}(a|s)}{\pi_{\theta_{n,1}}(a|s)} \in \left( e^{-\frac{\sqrt{2L}}{\lambda}}, \ e^{\frac{\sqrt{2L}}{\lambda}} \right).
$$

Now we choose

$$
\lambda = \lambda_0 := \min \left\{ \frac{\sqrt{2L}}{-\ln(1-\delta_0)}, \frac{\sqrt{2L}}{\ln(1+\delta_0)} \right\},
$$

which implies

$$
\frac{\pi_{\theta_{n,k}}(a|s)}{\pi_{\theta_{n,1}}(a|s)} \in \left( e^{-\frac{\sqrt{2L}}{\lambda_0}}, \ e^{\frac{\sqrt{2L}}{\lambda_0}} \right) \subset (1 - \delta_0, 1 + \delta_0),
$$

which then implies that $B_{n,k}^{(s,a)}$ happens. As $C_{n,k,\lambda_0}^{(s,a)} \subset B_{n,k}^{(s,a)}$, we have $\Omega / B_{n,k}^{(s,a)} \subset \Omega / C_{n,k,\lambda_0}^{(s,a)}$, $\mathbf{1}_{\Omega / B_{n,k}^{(s,a)}} \leq \mathbf{1}_{\Omega / C_{n,k,\lambda_0}^{(s,a)}}$. Substituting it into Equation (8) yields

$$
\|X_{n,k}\| \leq \tilde{\mathbb{E}}_{\pi_{\theta_{n,1}}} \left( \mathbf{1}_{\Omega / C_{n,k,\lambda_0}^{(s,a)}} \cdot \frac{\|\nabla \pi_{\theta_{n,1}}(a|s)\|}{\pi_{\theta_{n,1}}(a|s)} \cdot \left| \tilde{A}_{\pi_{\theta_{n,1}}}(s,a) \right| \right).
$$

We will now calculate the conditional expectation of $\|X_{n,k}\|$ on $\mathcal{F}_{n-1}$. Noting $\mathbf{1}_{\Omega / C_{n,k,\lambda_0}^{(s,a)}} \in \mathcal{F}_{n-1}$, we have

$$
\begin{aligned}
\mathbb{E} \left( \|X_{n,k}\| \big| \mathcal{F}_{n-1} \right) &\leq \mathbb{E} \left( \tilde{\mathbb{E}}_{\pi_{\theta_{n,1}}} \left( \mathbf{1}_{\Omega / C_{n,k,\lambda_0}^{(s,a)}} \frac{\|\nabla \pi_{\theta_{n,1}}(a|s)\|}{\pi_{\theta_{n,1}}(a|s)} \left| \tilde{A}_{\pi_{\theta_{n,1}}}(s,a) \right| \right) \Big| \mathcal{F}_{n-1} \right) \\
&\leq \mathbb{E}_{\pi_{\theta_{n,1}}} \left( \mathbf{1}_{\Omega / C_{n,k,\lambda_0}^{(s,a)}} \cdot \frac{\|\nabla \pi_{\theta_{n,1}}(a|s)\|}{\pi_{\theta_{n,1}}(a|s)} \cdot \left| \tilde{A}_{\pi_{\theta_{n,1}}}(s,a) \right| \right) + \sqrt{|\mathcal{A}|} R_{\max} \phi_n.
\end{aligned}
$$

Noting

$$
\begin{aligned}
&\mathbb{E}_{\pi_{\theta_{n,1}}} \left( \mathbf{1}_{\Omega / C_{n,k,\lambda_0}^{(s,a)}} \cdot \frac{\|\nabla \pi_{\theta_{n,1}}(a|s)\|}{\pi_{\theta_{n,1}}(a|s)} \cdot \left| \tilde{A}_{\pi_{\theta_{n,1}}}(s,a) \right| \right) \\
&\leq \sqrt{2L} \, \mathbb{E}_{\pi_{\theta_{n,1}}} \left( \mathbf{1}_{\Omega / C_{n,k,\lambda_0}^{(s,a)}} \cdot \frac{1}{\sqrt{\pi_{\theta_{n,1}}(a|s)}} \cdot \left| \tilde{A}_{\pi_{\theta_{n,1}}}(s,a) \right| \right) \\
&\leq |\mathcal{A}| \sqrt{2L} R_{\max} \, \mathbb{E}_{s \sim \pi_{\theta_{n,1}}} \left( \mathbf{1}_{\Omega / C_{n,k,\lambda_0}^{(s,a)}} \cdot \sqrt{\pi_{\theta_{n,1}}(a|s)} \right) \\
&\leq \sqrt{2L}(\sqrt{2L} + \lambda_0)\sqrt{1 + \delta_0} R_{\max}^3 |\mathcal{A}| \left( \sum_{i=1}^{k-1} \epsilon_{n,i} \right) \tilde{\mathbb{E}}_{\pi_{\theta_{n-1,1}}} \left( \frac{1}{\sqrt{\pi_{\theta_{n-1,1}}(a|s)}} \right),
\end{aligned}
$$

we get

$$
\begin{aligned}
\mathbb{E} \left( \|X_{n,k}\| \big| \mathcal{F}_{n-1} \right) &\leq \mathbb{E} \left( \tilde{\mathbb{E}}_{\pi_{\theta_{n,1}}} \left( \mathbf{1}_{\Omega / C_{n,k,\lambda_0}^{(s,a)}} \frac{\|\nabla \pi_{\theta_{n,1}}(a|s)\|}{\pi_{\theta_{n,1}}(a|s)} \left| \tilde{A}_{\pi_{\theta_{n,1}}}(s,a) \right| \right) \Big| \mathcal{F}_{n-1} \right) \\
&\leq \sqrt{2L}(\sqrt{2L} + \lambda_0)\sqrt{1 + \delta_0} R_{\max}^3 |\mathcal{A}| \left( \sum_{i=1}^{k-1} \epsilon_{n,i} \right) \tilde{\mathbb{E}}_{\pi_{\theta_{n-1,1}}} \left( \frac{1}{\sqrt{\pi_{\theta_{n-1,1}}(a|s)}} \right) \\
&\quad + \sqrt{|\mathcal{A}|} R_{\max} \phi_n.
\end{aligned} \tag{10}
$$

Now we inspect the norm of the term $Y_n$ in Equation (7). We have

$$\|Y_{n,k}\| = \left\| \tilde{\mathbb{E}}_{\pi_{\theta_{n,1}}} \left( \mathbf{1}_{B_{n,k}^{(s,a)}} \frac{\nabla \pi_{\theta_{n,k}}(a|s) - \nabla \pi_{\theta_{n,1}}(a|s)}{\pi_{\theta_{n,1}}(a|s)} \cdot \tilde{A}_{\pi_{\theta_{n,1}}}(s,a) \right) \right\|$$

$$\leq \tilde{\mathbb{E}}_{\pi_{\theta_{n,1}}} \left( \frac{\|\nabla \pi_{\theta_{n,k}}(a|s) - \nabla \pi_{\theta_{n,1}}(a|s)\|}{\pi_{\theta_{n,1}}(a|s)} \cdot \left| \tilde{A}_{\pi_{\theta_{n,1}}}(s,a) \right| \right)$$

$$\leq \tilde{\mathbb{E}}_{\pi_{\theta_{n,1}}} \left( \frac{L}{\pi_{\theta_{n-1}}(a|s)} \cdot \left( \sum_{t=1}^{k} \|\theta_{n,t} - \theta_{n,1}\| \right) \cdot \left| \tilde{A}_{\pi_{\theta_{n,1}}}(s,a) \right| \right)$$

$$\leq R_{\max}^3 L \left( \sum_{i=1}^{k-1} \epsilon_{n,i} \right) \tilde{\mathbb{E}}_{\pi_{\theta_{n-1,1}}} \left( \frac{\sqrt{2L(1+\delta_0)}}{\sqrt{\pi_{\theta_{n-1,1}}(a|s)}} \right) \cdot \tilde{\mathbb{E}}_{\pi_{\theta_{n,1}}} \left( \frac{1}{\pi_{\theta_{n,1}}(a|s)} \right).$$

Because

$$\tilde{\mathbb{E}}_{\pi_{\theta_{n-1,1}}} \left( \frac{\sqrt{2L(1+\delta_0)}}{\sqrt{\pi_{\theta_{n-1,1}}(a|s)}} \right) \in \mathcal{F}_{n-1},$$

we have

$$\mathbb{E} \left( \|Y_n\| \big| \mathcal{F}_{n-1} \right) \leq L \sqrt{2L(1+\delta_0)} R_{\max}^3 |\mathcal{A}| \left( \sum_{i=1}^{k-1} \epsilon_{n,i} \right) \tilde{\mathbb{E}}_{\pi_{\theta_{n-1,1}}} \left( \frac{1}{\sqrt{\pi_{\theta_{n-1,1}}(a|s)}} \right) \tag{11}$$

$$+ \sqrt{|\mathcal{A}|} R_{\max} \phi_n.$$

By the Lipschitz continuity of $\nabla \pi$, for any $\theta_1, \theta_2 \in \mathbb{R}^b$, there is

$$\|\nabla V(\theta_1) - \nabla V(\theta_2)\| \leq \left( \frac{|\mathcal{A}| R_{\max} L}{(1-\gamma)^2} + \frac{(1+\gamma)|\mathcal{A}| R_{\max} \cdot \sqrt{2L}}{(1-\gamma)^3} \right) \cdot \|\theta_1 - \theta_2\|.$$

For notational convenience, we assign above coefficient as $\mathcal{L} := \frac{|\mathcal{A}| R_{\max} L}{(1-\gamma)^2} + \frac{(1+\gamma)|\mathcal{A}| R_{\max} \cdot \sqrt{2L}}{(1-\gamma)^3}$. Then the discrete difference $V^* - V(\theta_{n+1,1}) - (V^* - V(\theta_{n,1}))$ of $V(\theta_{n,1})$ can be expanded as

$$V^* - V(\theta_{n+1,1}) - (V^* - V(\theta_{n,1})) \leq \nabla V(\theta_{n,1})^\top (\theta_{n+1,1} - \theta_{n,1}) + \mathcal{L} \|\theta_{n+1,1} - \theta_{n,1}\|^2$$

$$= \nabla V(\theta_{n-1})^\top (\theta_{n+1,1} - \theta_{n,1}) + \left( \nabla V(\theta_{n,1}) - \nabla V(\theta_{n-1}) \right)^\top (\theta_{n+1,1} - \theta_{n,1}) \tag{12}$$

$$+ \mathcal{L} \|\theta_{n+1,1} - \theta_{n,1}\|^2.$$

We substitute Equation (2) and Equation (7) into Equation (12) and obtain

$$V(\theta_{n+1,1}) - V(\theta_{n,1}) \leq - \left( \sum_{k=1}^{T-1} \epsilon_{n,k} \right) \nabla V(\theta_{n-1,1})^\top \tilde{\mathbb{E}}_{\pi_{\theta_{n,1}}} \left( \frac{\nabla \pi_{\theta_{n,1}}(a|s)}{\pi_{\theta_{n,1}}(a|s)} \cdot \tilde{A}_{\pi_{\theta_{n,1}}}(s,a) \right)$$

$$- \nabla V(\theta_{n-1})^\top \sum_{k=1}^{T-1} \epsilon_{n,k} X_{n,k} - \nabla V(\theta_{n-1})^\top \sum_{k=1}^{T-1} \epsilon_{n,k} Y_{n,k} \tag{13}$$

$$+ \mathcal{L} \|\theta_{n,1} - \theta_{n-1,1}\| \cdot \|\theta_{n+1,1} - \theta_{n,1}\| + \mathcal{L} \|\theta_{n+1,1} - \theta_{n,1}\|^2.$$

Then substituting Equation (10) and Equation (11) into Equation (13) will acquire

$$V^* - V(\theta_{n+1,1}) - (V^* - V(\theta_{n,1})) \leq - \left( \sum_{k=1}^{T-1} \epsilon_{n,k} \right) \|\nabla V(\theta_{n,1})\|^2$$

$$+ 4\sqrt{1+\delta_0} R_{\max}^2 \sqrt{|\mathcal{A}|} \mathcal{L} \left( \sum_{k=1}^{T-1} \epsilon_{n,k} \right) \left( \sum_{k=1}^{T-1} \epsilon_{n+1,k} \right) \|\nabla V(\theta_{n,1})\|^2 \tag{14}$$

$$+ 2\sqrt{|\mathcal{A}|} \mathcal{L} \left( \phi_n + O\left( \sum_{k=1}^{T-1} \epsilon_{n,k} \right) \right) \cdot \left( \sum_{k=1}^{T-1} \epsilon_{n,k} \right).$$

The lemma follows. $\qquad\square$

## A.3 PROOF OF THEOREM 3.2

*Proof.* Through Lemma 3.1, we know $\forall\, n > t$, there is

$$
\mathbb{E}(V^* - V(\theta_{n+1,1})) - \mathbb{E}((V^* - V(\theta_{n,1}))) \le -\bigg(\sum_{k=1}^{T-1} \epsilon_{n,k}\bigg) \mathbb{E}\left\|\nabla V(\theta_{n,1})\right\|^2
$$
$$
+ 4\sqrt{1+\delta_0}R_{\max}^2\sqrt{|\mathcal{A}|}\mathcal{L}\bigg(\sum_{k=1}^{T-1}\epsilon_{n,k}\bigg)\bigg(\sum_{k=1}^{T-1}\epsilon_{n+1,k}\bigg)\mathbb{E}\left\|\nabla V(\theta_{n,1})\right\|^2 \tag{15}
$$
$$
+ 2\sqrt{|\mathcal{A}|}\mathcal{L}\cdot O\bigg(\sum_{k=1}^{T-1}\epsilon_{n,k}^2\bigg).
$$

By inspecting the asymptotic order, we conclude the existence of an $N_0$ such that for $n > N_0$, we have

$$
\bigg(\sum_{k=1}^{T-1}\epsilon_{n+1,k}\bigg)\mathbb{E}\left\|\nabla V(\theta_{n,1})\right\|^2 > 8\sqrt{1+\delta_0}R_{\max}^2\sqrt{|\mathcal{A}|}\mathcal{L}\bigg(\sum_{k=1}^{T-1}\epsilon_{n,k}\bigg)\bigg(\sum_{k=1}^{T-1}\epsilon_{n+1,k}\bigg)\mathbb{E}\left\|\nabla V(\theta_{n,1})\right\|^2.
$$

Note that here $N_0$ depends only on the parameters $\delta_0$, $R_{\max}$, $|\mathcal{A}|$, $\mathcal{L}$ and the learning rate $\epsilon_{n,k}$. Substituting the above inequity into (15), we know that when $n > i_0 := \max\{N_0, t\}$, there is

$$
\mathbb{E}(V^* - V(\theta_{n+1,1})) - \mathbb{E}((V^* - V(\theta_{n,1}))) \le -\frac{1}{2}\bigg(\sum_{k=1}^{T-1}\epsilon_{n,k}\bigg)\mathbb{E}\left\|\nabla V(\theta_{n,1})\right\|^2
$$
$$
+ 2\sqrt{|\mathcal{A}|}\mathcal{L}\cdot O\bigg(\sum_{k=1}^{T-1}\epsilon_{n,k}^2\bigg). \tag{16}
$$

Summing over Equation (16) from $i_0$ to $t$, and noticing the condition $\sum_{n=1}^{+\infty}\big(\sum_{k=1}^{T-1}\epsilon_{n,k}^2\big) < +\infty$, we obtain

$$
\frac{1}{2}\sum_{n=i_0}^{t}\bigg(\sum_{k=1}^{T-1}\epsilon_{n,k}\bigg)\mathbb{E}\left\|\nabla V(\theta_{n,1})\right\|^2 < +\infty,
$$

which implies

$$
\frac{1}{2}\sum_{n=i_0}^{t}\bigg(\sum_{k=1}^{T-1}\epsilon_{n,k}\bigg)\left\|\nabla V(\theta_{n,1})\right\|^2 < +\infty \;\; a.s.. \tag{17}
$$

The condition $\sum_{n=1}^{+\infty}\big(\sum_{k=1}^{T-1}\epsilon_{n,k}\big) = +\infty$ implies $\liminf_{n\to+\infty}\left\|\nabla V(\theta_{n,1})\right\| = 0\; a.s.$, which means that for any $\delta > 0$, there exists a subsequence $\{q_n\}_{n=1}^{+\infty}$ such that $\left\|\nabla g(\theta_{q_n})\right\| < \delta$. We find all the boundary points in the sequence $\{q_n\}_{n=1}^{+\infty}$ and denote them as $\{p_n\}_{n=1}^{+\infty}$. They satisfy $\left\|\nabla V(\theta_{p_{2k-1},1})\right\| \le \delta$, $\left\|\nabla V(\theta_{p_{2k-1}+1,1})\right\| > \delta$, $\left\|\nabla V(\theta_{p_{2k}-1,1})\right\| > \delta$, $\left\|\nabla V(\theta_{p_{2k},1})\right\| \le \delta$. If $\{p_n\}_{n=1}^{+\infty}$ contains at most finitely many elements, there exists an $n_0 > 0$, such that for any $n > n_0$, $\left\|\nabla V(\theta)\right\| < \delta$. If $\{p_n\}_{n=1}^{+\infty}$ has infinitely many elements, we find

$$
\sup_{p_{2k-1}<n\le p_{2k}}\left\|\nabla V(\theta_{n,1})\right\| \le \sum_{n=p_{2k-1}}^{p_{2k}-1}\big|\left\|\nabla V(\theta_{n+1,1})\right\| - \left\|\nabla V(\theta_{n,1})\right\|\big| \le \mathcal{L}\sum_{n=p_{2k-1}}^{p_{2k}-1}\left\|\theta_{n+1,1}-\theta_{n,1}\right\|.
$$

By substituting the expression of PPO-Clip (2) into the above inequality, and combining Lemma A.2, we obtain

$$
\sup_{p_{2k-1}<n\le p_{2k}}\left\|\nabla V(\theta_{n,1})\right\| \le \mathcal{L}\sqrt{2|\mathcal{A}|L(1+\delta_0)}\sum_{n=p_{2k-1}}^{p_{2k}-1}\sum_{k=1}^{T}\epsilon_{n,k}
$$
$$
\le \frac{\mathcal{L}R_{\max}\sqrt{2|\mathcal{A}|L(1+\delta_0)}}{\delta}\sum_{n=p_{2k-1}}^{p_{2k}-1}\bigg(\sum_{k=1}^{T}\epsilon_{n,k}\bigg)\left\|\nabla V(\theta_{n,1})\right\|^2.
$$

According to Equation (17), there exists $n_1 > 0$, such that for any $p_{2k-1} > n_1$,

$$\frac{\mathcal{L}\sqrt{2|\mathcal{A}|L(1+\delta_0)}}{\delta} \sum_{n=p_{2k-1}}^{p_{2k}-1} \left( \sum_{k=1}^{T} \epsilon_{n,k} \right) \|\nabla V(\theta_{n,1})\|^2 < \delta.$$

Subsequently, for any $n > \max\{i_0, n_1\}$, there is

$$\|\nabla V(\theta_{n,1})\| \leq \|\nabla V(\theta_{p_{2k-1},1})\| + \sup_{p_{2k-1} < n \leq p_{2k}} \|\nabla V(\theta_{n,1})\| < 2\delta,$$

which indicates that for any $n > \max\{i_0, n_0, n_1\}$, there is $\|\nabla V(\theta_{n,1})\| < 2\delta$. By the arbitrariness of $\delta$, we have

$$\lim_{n \to +\infty} \|\nabla V(\theta_{n,1})\| = 0 \ a.s..$$

Therefore, for any $\theta_{n,1}$, we have $\|\nabla V(\theta_{n,1})\|^2 < R_{\max}^2 2|\mathcal{A}|L(1+\delta_0)$. Through the Lebesgue's dominated convergence theorem, we obtain

$$\lim_{n \to +\infty} \mathbb{E} \|\nabla V(\theta_{n,1})\|^2 = 0$$

as we desired. □

