# OpenReview forum: "On Stationary Point Convergence of PPO-Clip"
_ICLR.cc/2024/Conference — ICLR 2024 poster_

### Official Review · Reviewer_v9Uz · 2023-10-22

**Soundness:** 3 good
**Presentation:** 3 good
**Contribution:** 2 fair
**Rating:** 3
**Confidence:** 4

**Summary:**

The paper provides a theoretical analysis of the convergence of a variant of the popular PPO algorithm where the objective function is clipped.

**Strengths:**

The paper provides formal guarantees for the convergence of the clipping variant of PPO that has been used widely by reinforcement learning practitioners. This probably adds some reassurance that that this is a sound RL method to use.

**Weaknesses:**

The paper appears to be a good contribution to the filed of RL, but has little, if anything, to do with representation learning. ICLR'24 might not be the best venue for this paper.

Some minor typos:
Last paragraph on page 1: "this analysis rely" -> "this analysis relies"
Same place: "no longer involve" -> "no longer involves"

**Questions:**

Can you think of a possible impact your result can have on the field of representation learning?

---

> ### Author Response · Authors · 2023-11-16
> **Author rebuttal**
>
> We thank the reviewer for providing feedback. We are glad to hear that the reviewer acknowledge that "The paper appears to be a good contribution to the filed of RL".
>
> We re-examined the Call For Papers of ICLR 2024, where it states
>
> > We consider a broad range of subject areas including feature learning, metric learning, compositional modeling, structured prediction, reinforcement learning, uncertainty quantification and issues regarding large-scale learning and non-convex optimization, as well as applications in vision, audio, speech, language, music, robotics, games, healthcare, biology, sustainability, economics, ethical considerations in ML, and others.
>
> Here, reinforcement learning is clearly one of the topics that ICLR 2024 is interested in. Additionally, in "A non-exhaustive list of relevant topics", the second topic is "representation learning for planning and reinforcement learning". We also searched ICLR 2022 accepted papers, where a simple query of "reinforcement learning" returns 68 papers, and querying "policy optimization" returns 7 papers. We therefore believe that manuscripts that have a good contribution to reinforcement learning should be considered by ICLR 2024.
>
> For possible impacts, we understand that it is hard to imagine the immediate use cases of the work, because our analysis is on the original version of PPO-Clip and does not introduce new algorithms. Nevertheless, this does not mean a lack of implication. One example is our analysis provides an understanding toward the clip operator, which is not typically used in optimization. Our analysis on the events $B_{n,k}$ and $C_{n,k}$ characterizes the effectiveness of the clip and also how often the clip is expected to happen in different stages of the optimization. Another example is our setting of $T$, which denotes the number of steps that the same old reference policy is used. If $T$ is too large, then the clip happens too often, and the gradient becomes zero too often which makes the update less efficient. If $T$ is small, then the samples are less effectively utilized, and the algorithm will take more time interacting with the environment. There will be a tradeoff between these two effects. The impact of this work will be its long-term implication, where a better understanding of our algorithm could involve new variants of the algorithm to be developed. Considering the popularity of the algorithm, bringing up new understanding is valuable.
>
> We thank the reviewer for pointing out several grammatical errors and we have revised them in the updated manuscript.
>
> [1] ICLR 2024 - Call For Papers https://iclr.cc/Conferences/2024/CallForPapers
>
> [2] ICLR 2022 Accepted Papers https://iclr.cc/virtual/2022/papers.html

---

> > ### Comment · Reviewer_v9Uz · 2023-11-22
> > **Response appreciated**
> >
> > Thank you for addressing my comments and answering my questions.

---

> ### Author Response · Authors · 2023-11-22
> **Have we addressed your concerns?**
>
> We thank the reviewer again for providing feedback on our paper. As the discussion period is coming to a close, we would like to know if we have resolved your concerns expressed in the original reviews, especially the comments that we believe might contradict the ICLR 2024 CFP. We remain open to any further feedback and are committed to making additional improvements if needed. If you find that these concerns have been resolved, we would be grateful if you would consider reflecting this in your rating of our paper.

---

> ### Author Response · Authors · 2023-11-22
> **Thank you**
>
> We thank the reviewer for the feedback and the further response. In case the concerns in the original review have been addressed and the questions have been answered, could the reviewer reflect this change in the rating of our manuscript? In case any concern remains, we are committed to further discussing with the reviewer and making updates to improve our manuscript.

---

### Official Review · Reviewer_wVXF · 2023-10-29

**Soundness:** 4 excellent
**Presentation:** 2 fair
**Contribution:** 2 fair
**Rating:** 6
**Confidence:** 2

**Summary:**

This work takes a closer look at the theories behind the clipped surrogate objective of PPO (Proximal Policy Optimization). The authors provide a comprehensive analysis that proves the stationary point convergence of PPO-Clip and demonstrates the convergence rate.

**Strengths:**

The work is novel in that it investigates the theoretical convergence of PPO, whereas the field has overwhelmingly focused on the empirical performance and application of PPO (e.g., Dota 2 with PPO and ChatGPT, RLFH with PPO). This is partially because the clip operation is inherently non-smooth, thus posing challenges to empirical analysis. The authors' analysis seems sound and comprehensive; they also clearly listed out the necessary list of assumptions. The authors' Theorem 3.2. indicates "PPO-Clip has the same convergence property as the best current results available for policy gradient", which seems significant.

**Weaknesses:**

Please take this with a grain of salt, as I have primarily been using PPO under empirical settings. I struggle to understand how this work connects with the wider research community. What is the implication of this work? This work demonstrates PPO has stationary point convergence — can this property be used in some ways?

**Questions:**

> the unbounded score function makes the ratio of two policies arbitrarily large, even in the late stages of the optimization process.
Do the authors mean the ratio **could** become arbitrarily large?

---

> ### Author Response · Authors · 2023-11-16
> **Author rebuttal**
>
> We thank the reviewer for the feedback. We are glad to learn that the reviewer acknowledges our novelty and the challenges involved in the analysis. Indeed, there are few analysis that directly tackles the vanilla version of the PPO with the clip surrogate objective. We are pleased to bring up the convergence characterizations of PPO-Clip, under a minimum set of assumptions.
>
> We fully agree with the reviewer that it is hard to imagine the immediate use cases of the work. The primary cause is that our analysis is on the original version of PPO-Clip and does not introduce new algorithms. Nevertheless, this does not mean a lack of implication. One example is our analysis provides an understanding toward the clip operator, which is not typically used in optimization. Our analysis on the events $B_{n,k}$ and $C_{n,k}$ characterizes the effectiveness of the clip and also how often the clip is expected to happen in different stages of the optimization. Another example is our setting of $T$, which denotes the number of steps that the same old reference policy is used. If $T$ is too large, then the clip happens too often, and the gradient becomes zero too often which makes the update less efficient. If $T$ is small, then the samples are less effectively utilized, and the algorithm will take more time interacting with the environment. There will be a tradeoff between these two effects.
>
> With this level of popularity of PPO in particular (e.g. 5,500 search results on GitHub), we believe it is relevant to overcome the technical challenges, and bring the understanding of PPO up to the community. Such knowledge could result in a more long-term implication that invokes more new algorithms to be proposed, which could be more valuable than proposing a single variant alone.
>
> The reviewer's comment on the ratio of two policies is well taken and the corresponding fix has been made in the updated pdf.

---

> > ### Comment · Reviewer_wVXF · 2023-11-22
> > **Reply**
> >
> > Thank you for the response.

---

> > > ### Author Response · Authors · 2023-11-22
> > > **Thank you**
> > >
> > > We thank the reviewer again for your time and effort in reviewing our paper. We are glad that our novelty in the analysis is recognized and it is nice to have the opportunity to elaborate potential implications. We very much enjoyed the discussion with the reviewer.

---

### Official Review · Reviewer_dSqC · 2023-11-01

**Soundness:** 3 good
**Presentation:** 3 good
**Contribution:** 3 good
**Rating:** 8
**Confidence:** 4

**Summary:**

This paper presents theoretical analysis of convergence of PPO algorithm using clipping and version of two-time scale method, i.e., updating policy parameter with particular period. Analysis of PPO-clip is difficult due to non-smoothness of clipping operator and involves ratio of policies. Theorem 3.3 provides best iterate and averaged iterate convergence in terms of $||\nabla V(\theta_{n,1})||\to 0$.

**Strengths:**

- The paper tackles important question regrading theoretical analysis of practical implementation of PPO algorithm. The aim of the paper is clear and simple.

- The analysis seems to be novel. The authors use two-time scale method to overcome the difficulty to deal with ratio of the policy and constructing particular set of events enables to derive recursive inequalities to bound the norm of gradient of the clipped loss function.

**Weaknesses:**

- Even though the aim of this paper is to tackle theoretical analysis of practical implementation of PPO, still there is some gap. Using a inner and outer loop style update seems to be far from practical implementation.
-  The estimate for advantage function, $\phi_n$ can be estimated with parameterized value network (e.g. neural network). However, I believe extending the proof to Actor-Critic setting is non-trivial. Hence, I believe the proof is setting restricted to Monte-Carlo setting.
- Assumption 3.4. restricts the generality of $T$ and the learning rate. Providing simple examples on the learning rate, and $T$ would be helpful to understand the conditions about Assumption 3.4. For example, can we use $\frac{1}{k}$ or $\frac{1}{\sqrt{k}}$ as the step-size?

**Questions:**

- Above the paragraph of Step 1, what does it mean to have similar recursive inequality like policy gradient? Please provide more details.

- In Step 1, how does the estimated clipped PPO loss have gradient? Should it be sub-gradient due to the non-smoothness of the clipping operator? Please provide more clarifications.

- What is the intuitive meaning of $X_{n,k}$ and $Y_{n,k}$? The motivation of decomposition of error term  into $X_{n,k}$ and $Y_{n,k}$ is not really clear

- The introduction of $C_{n,k}$ in Step 2 is quite abrupt. Please provide more details.

- In Step 2, what is the meaning of bound of $\mathbb{E}[||X_k|| \mathcal{F} ]$? Depending on $\frac{1}{\sqrt{\pi_{\theta_{n-1,1}}(s,a)}}$ seems to be problematic. How is this term compensated? In deriving (14) where did $\frac{1}{\sqrt{\pi_{\theta_{n-1,1}(a\mid s)}}}$ go?

- I think there is typo in (13). $\nabla V(\theta_{n-1})$ should be $\nabla V(\theta_{n-1,1})$.

---

> ### Author Response · Authors · 2023-11-16
> **Author rebuttal (Part 1)**
>
> We thank the reviewer for the feedback. We are glad to have our aim and our technical novelty acknowledged. One thing we wanted to clarify is that our formulation in Section 3.1 includes the single-timescale setting (which is what the vanilla implementation of PPO did in OpenAI Baselines). **Our results hold for both the single-timescale version (the vanilla version) and the two-timescale version (the version that the optimizer is run for $T$ times per sample, which is also commonly used)**.
>
> We now provide detailed responses to the reviewer's questions and comments
>
> > 1. Even though the aim of this paper is to tackle theoretical analysis of practical implementation of PPO, still there is some gap. Using a inner and outer loop style update seems to be far from practical implementation.
>
> Fortunately, our results hold for both the single-timescale version and the two-timescale version of PPO. We believe this is a strength rather than a weakness.
>
> > 2. The estimate for advantage function, $\phi_n$ can be estimated with parameterized value network (e.g. neural network). However, I believe extending the proof to Actor-Critic setting is non-trivial. Hence, I believe the proof is setting restricted to Monte-Carlo setting.
>
> We fully agree with the reviewer. We unify the estimation error of the advantage, including the sampling error, into $\phi_{n}$. Indeed, characterizing this error is very challenging. It involves estimating the fitting error of the neural network, which is an important problem in neural network theory. To the best of our knowledge, this problem is solved for some specific networks, such as over-parameterized networks and linear neural networks [2, 3]. Extending the proof to Actor-Critic will result in the $\phi_n$ variable unbounded.
>
> > 3. Assumption 3.4 restricts the generality of $T$ and the learning rate. Providing simple examples on the learning rate, and $T$ would be helpful to understand the conditions about Assumption 3.4. For example, can we use $\frac{1}{k}$ or $\frac{1}{\sqrt{k}}$ as the step-size?
>
> We first wanted to explain that $T$ is a small value, something between $1$ to $10$ in practice. It is the number of off-policy PPO updates on the same reference old policy. Our notation of calling it "T" might results in some confusion that it sounds like something large.
>
> Assumption 3.4 is only related to $k$, which approaches to infinity, and is not related to $T$, which is a small constant. The assumption is actually quite weak, which only requires the learning rate to not decay too quickly. Examples are $\epsilon_{n,k}=1/\sqrt{n}$, $\epsilon_{n,k}=1/n$, and $\epsilon_{n,k}=1/n\ln n$. Examples that do not satisfy our assumption are $\epsilon_{n,k}=1/n^2$, which decreases too fast and causes the value function update to stop before reaching a stationary point.
>
> > 4. Above the paragraph of Step 1, what does it mean to have similar recursive inequality like policy gradient? Please provide more details.
>
> For the PG algorithm, i.e., $$\theta_{n+1}=\theta_{n}+\epsilon_{n}\hat{\nabla} V_{H}(\theta_{n}),$$ we can obtain the following recursive inequality
> $$E(V^*-V(\theta_{n+1})|F_n)\le V^*-V(\theta_n)-\epsilon_n\hat{\nabla}(\theta_{n})^{\top}\nabla V(\theta_{n})+\frac{{L}\epsilon_n^2}{2}\\|\hat{\nabla}V(\theta_{n})\\|^{2}.$$
> This inequality is crucial for the convergence of the PG algorithm. Therefore, a very natural approach to prove the convergence of the PPO algorithm is to construct a similar inequality to the one mentioned above, and thereby establish the convergence of PPO.
>
> > 5. In Step 1, how does the estimated clipped PPO loss have gradient? Should it be sub-gradient due to the non-smoothness of the clipping operator? Please provide more clarifications. What is the intuitive meaning of $X_{n,k}$ and $Y_{n,k}$? The motivation of decomposition of error term into $X_{n,k}$ and $Y_{n,k}$ is not really clear.
>
> The gradient of the clipped variable is defined as normal gradient when it is unclipped, and zero when it is clipped. This is the same as what the auto-gradient will implement in practice. The only caveat thing in analysis is the "boundary" points, where the ratio is exactly $1+\delta_0$ or $1-\delta_0$. We set the gradient of these points as zero. In fact, we can set the gradient to any value in the subgradient and the analysis still holds, because those boundary points set has a zero measure.
>
> For the events, $X_{n,k}$ and $Y_{n,k}$ represent the errors of the object with the clip operation and the object without the clip operation, respectively. Since the remaining part after removing the clip operation (the first term on the right side of the inequality in Step 1) is a standard policy gradient, by bounding these two error terms we could obtain the final result.

---

> ### Author Response · Authors · 2023-11-16
> **Author rebuttal (Part 2)**
>
> > 6. The introduction of $C_{n,k}$ in Step 2 is quite abrupt. Please provide more details.
>
> The purpose of introducing this event is to estimate the range in which the clip operation holds. We have scaled down the range of the clip operation to an event that is only related to $\pi_{\theta_{n,1}}$ and $\pi_{\theta_{n-1,1}}$. In this way, we have transformed a double-loop problem into a single-loop problem. In other words, $C_{n,k}$ is a relaxation of $B_{n,k}$.
>
> > 7. In Step 2, what is the meaning of bound of $\mathbb{E}[||X_k|| \mathcal{F} ]$? Depending on $\frac{1}{\sqrt{\pi_{\theta_{n-1,1}}(s,a)}}$ seems to be problematic. How is this term compensated? In deriving (14) where did $\frac{1}{\sqrt{\pi_{\theta_{n-1,1}(a\mid s)}}}$ go?
>
> We thank the reviewer for pointing this out. We did made a typo in this term in the original version of the appendix. In the revised version (updated pdf), we have corrected this typo. We additionally added the detailed derivation to clarify how the term is obtained. Please refer to lines 17-27 on page 14 of the revised appendix for the updated information.
>
> >8. I think there is typo in (13). $\nabla V(\theta_{n-1})$ should be $\nabla V(\theta_{n-1,1})$.
>
> We thank the reviewer for pointing this out. We corrected this in the revised version.
>
> [1] OpenAI Baselines https://github.com/openai/baselines
>
> [2] Du, Simon S., Xiyu Zhai, Barnabas Poczos, and Aarti Singh. "Gradient descent provably optimizes over-parameterized neural networks." arXiv preprint arXiv:1810.02054 (2018).
>
> [3] Baldi, Pierre, and Kurt Hornik. "Neural networks and principal component analysis: Learning from examples without local minima." Neural networks 2, no. 1 (1989): 53-58.

---

> > ### Comment · Reviewer_dSqC · 2023-11-17
> > **Response to authors**
> >
> > Thank you for the detailed response, and most of the concerns have been addressed. I have increased my score from 6 to 8.

---

> ### Author Response · Authors · 2023-11-22
> **Thank you**
>
> We thank the reviewer again for your time and effort in reviewing our paper. Your review helps us a lot in improving the manuscript. We are glad to know that the reviewer recognized our contributions and we very much enjoyed the discussion with the reviewer. As the discussion period is coming to a close, we would like to note that we remain open to any further feedback and are committed to making additional improvements.

---

### Meta-Review · Area_Chair_4Qi7 · 2023-12-09

**Metareview:**

This paper presents theoretical analysis of convergence of PPO algorithm that uses clipping and the two-time scale method. Analysis in this work appear to be novel with the use of (1) two-time scale method that technical overcomes the technical difficulty related to policy ratio, and  (2) recursive inequalities to bound the norm of gradient of the clipped loss function. The paper studies an important theoretical analysis of the popular PPO algorithm, which provides a pivotal theoretical bound to many RL approaches that relies on PPO.

Reviews submitted by v9Uz is omitted due to its brevity and quality.

**Justification For Why Not Higher Score:**

Reviewers still have questions regarding some technical details such as the theoretical guarantees of more general learning rates, and presentation of the paper requires more clarification/improvement,

**Justification For Why Not Lower Score:**

Paper has significant novel contribution on proving convergence of the PPO algorithm. Theoretical contributions appear to be relevant to a wide RL research community.

---

### Decision · Program_Chairs · 2024-01-16

Accept (poster)